# Carvedilol suppresses ryanodine receptor-dependent Ca²⁺ bursts in human neurons bearing *PSEN1* variants found in early onset Alzheimer's disease

**Atsushi Hori[1], Haruka Inaba[2], Takashi Hato[3], Kimie Tanaka[2], Shoichi Sato[4], Mizuho Okamoto[2,5], Yuna Horiuchi[1], Faith Jessica Paran[2], Yoko Tabe[2], Shusuke Mori[6], Corina Rosales[7,8], Wado Akamatsu[5], Takashi Murayama[9], Nagomi Kurebayashi[9], Takashi Sakurai[9], Tomohiko Ai[2,6,10]\*, Takashi Miida[1,2]**

1 Department of Clinical Laboratory Technology, Faculty of Medical Science, Juntendo University, Chiba, Japan, 2 Department of Clinical Laboratory Medicine, Juntendo University Graduate School of Medicine, Tokyo, Japan, 3 Division of Nephrology, Department of Medicine, Indiana University School of Medicine, Indianapolis, Indiana, United States of America, 4 Department of Clinical Engineering, Faculty of Medical Science, Juntendo University, Chiba, Japan, 5 Center for Genomic and Regenerative Medicine, School of Medicine, Juntendo University, Tokyo, Japan, 6 Department of Acute Care and Disaster Medicine, Tokyo Medical and Dental University, Tokyo, Japan, 7 Center for Bioenergetics and the Department of Medicine, Houston Methodist Research Institute, Texas, United States of America, 8 Weill Cornell Medicine, New York, New York, United States of America, 9 Department of Pharmacology, Juntendo University School of Medicine, Tokyo, Japan, 10 Department of Clinical Laboratory Medicine, Juntendo University Urayasu Hospital, Chiba, Japan

\* t-ai@juntendo.ac.jp

**Data Availability Statement:** All relevant data are within the manuscript and its Supporting Information files.

## Abstract

Seizures are increasingly being recognized as the hallmark of Alzheimer's disease (AD). Neuronal hyperactivity can be a consequence of neuronal damage caused by abnormal amyloid β (Aß) depositions. However, it can also be a cell-autonomous phenomenon causing AD by Aß-independent mechanisms. Various studies using animal models have shown that Ca²⁺ is released from the endoplasmic reticulum (ER) via type 1 inositol triphosphate receptors (InsP₃R1s) and ryanodine receptors (RyRs). To investigate which is the main pathophysiological mechanism in human neurons, we measured Ca²⁺ signaling in neural cells derived from three early-onset AD patients harboring Presenilin-1 variants (*PSEN1* p. A246E, p.L286V, and p.M146L). Of these, it has been reported that PSEN1 p.A246E and p. L286V did not produce a significant amount of abnormal Aß. We found all *PSEN1*-mutant neurons, but not wild-type, caused abnormal Ca²⁺-bursts in a manner dependent on the calcium channel, Ryanodine Receptor 2 (RyR2). Indeed, carvedilol, an RyR2 inhibitor, and VK-II-86, an analog of carvedilol without the β-blocking effects, sufficiently eliminated the abnormal Ca²⁺ bursts. In contrast, Dantrolene, an inhibitor of RyR1 and RyR3, and Xestospongin c, an IP₃R inhibitor, did not attenuate the Ca²⁺-bursts. The Western blotting showed that RyR2 expression was not affected by *PSEN1* p.A246E, suggesting that the variant may activate the RyR2. The RNA-Seq data revealed that ER-stress responsive genes were increased, and mitochondrial Ca²⁺-transporter genes were decreased in PSEN1$_{A246E}$ cells compared to the WT neurons. Thus, we propose that aberrant Ca²⁺ signaling is a key link

**Funding:** Japan Society for the Promotion of Science (JSPS) grant 20K16532 (AH) JSPS grant 18K08900 and 21K08135 (TA).

**Competing interests:** The authors have declared that no competing interests exist.

between human pathogenic *PSEN1* variants and cell-intrinsic hyperactivity prior to deposition of abnormal Aß, offering prospects for the development of targeted prevention strategies for at-risk individuals.

## Introduction

Epileptic activities in patients with Alzheimer's disease (AD) have been gaining attentions due to the potential association with their abnormal cognitive behaviors [1, 2]. Specifically, seizures tend to appear more frequently in patients with early-stage AD than in elderly patients [1]. Indeed, in 2017, Lam et al. recorded subclinical seizure-like local neural hyper-excitations using intracranial catheters placed adjacent to the hippocampus in two human AD patients. The study showed that elimination of the subclinical seizures with levetiracetam, an anticonvulsant, improved the patient's abnormal cognitive behavior although the underlying mechanisms of the seizure-like signals remain unknown [3]. Therefore, several different types of anticonvulsants have been used to treat AD. However, there is no clear consensus on which medications are suitable for which patients' symptoms, such as epileptic amnesia and cognitive decline since the pathophysiological mechanisms of seizures in AD remains unelucidated [2].

To date, several hundreds of variants in the presenilin 1 gene (*PSEN1*) have been linked to early-onset familial AD [4, 5]. PSEN1 located in the membrane of the endoplasmic reticulum (ER) regulates intracellular Ca$^{2+}$ homeostasis [5]. Cysteines located in the 7th and 9th transmembrane portion of PSEN1 are essential to pass Ca$^{2+}$ in mouse models [6]. It has been reported that *PSEN1* variants found in AD produce abnormal amyloid β (Aß) and Tau proteins that damage neurons [7, 8]. However, recent studies revealed that not all pathogenic *PSEN1* variants produce abnormal Aß proteins [9], illustrating the importance of amyloid-independent mechanisms in the development of AD [10].

Also, intracellular Ca$^{2+}$-dysregulation has been observed in various AD animal models. In a classic murine AD model bearing a *PSEN1* p.M146V variant, glutamate provoked abnormal persistent Ca$^{2+}$ waves via the NMDA receptors, which may have an association with neurodegeneration [11]. Shilling et al. proposed that the type 1 inositol triphosphate receptor (InsP$_3$R1) is a main mediator to release Ca$^{2+}$ from the ER in the *PSEN1* p.M146V knock-in mouse model [12]. SanMatin et al. stated that Aβ oligomers promote ryanodine receptor (RyR)-mediated Ca$^{2+}$-release in primary hippocampal neurons of Sprague-Dawley embryos [13]. Yao et al. reported that limiting RyR open time via introducing a RyR2 variant, p. E4872Q, prevented AD-associated neuronal hyperexcitations in CA1-pyramidal neurons of the AD mouse model (5xFAD) despite an accumulation of Aß [14]. Schrank et al. generated a human neuronal model using iPS cells (iPSCs) obtained from an AD patient. They observed that production of abnormal Aß$_{42}$ and abnormal Ca$^{2+}$ were more prominent in the AD neurons than control cells, and dantrolene, an inhibitor of RyR1 and RyR3, eliminated both Aß$_{42}$ and Ca$^{2+}$ surges [15]. Furthermore, Gong et al. reported that variants of *RYR3* are associated with hypertension, diabetes, and AD [16].

In this study, we hypothesize that dysregulated Ca$^{2+}$ homeostasis and resultant neuronal hyperexcitability, independent from the abnormal Aß-theory, could be a discernable feature of early-onset AD associated with certain *PSEN1* variants. Accordingly, we analyzed intracellular Ca$^{2+}$ activities in iPS cell-derived neurons from multiple early-onset AD patients with pathogenic *PSEN1* variants that do not produce abnormal amyloid [9]. We found that all neuronal cells bearing *PSEN1* variants exhibited abnormal Ca$^{2+}$-waves, but not WT neurons. Also, the

RNA-Seq data showed that in the $PSEN1_{A246E}$ cells, ER stress responsive genes such as *EIF2S1* and *ATF4* were more expressed than the WT neurons. These data implicate a potential higher risk of subclinical seizures in this population.

## Materials and methods

### Cell culture and immunostaining reagents

All cells used in this study were commercially available cell lines purchased from Axol Bioscience (Cambridge, United Kingdom). This study protocol has been approved by the Institutional Review Board for Medical Research at the Juntendo University School of Medicine (IRB#E23-0210). Please contact them by the following email if necessary (E-mail: igakubu.rin-ri1@juntendo.ac.jp). Obtaining informed consents from the patients was not feasible since all cells were purchased from Axol Bioscience. Thus, the informed consents were waived by the IRB committee.

The reagents that were used in this study are listed as follows: ibidi GmbH, Gräfelfing, Germany: Polymer coverslip-bottom dishes (ib81156); chamber slide (ib80826); mounting medium with 4′,6-diamidino-2-phenylindole (DAPI) (ib50011). Axol Biosciences, Cambridge, UK: SureBond-XF (ax0053); Neural Maintenance Medium Supplement (ax0031); Neural Maintenance Basal Medium (ax0031); human FGF2 (ax0047); human EGF (ax0048); NeurOne Supplement A (ax0674a); NeurOne Supplement B (ax0674b). Merck, Darmstadt, Germany: polyornithine aqueous solution (P36555); bovine serum albumin (A2153); anti-β-tubulin III antibody (T8660); anti-microtubule-associated protein2 antibody (AB5622); 2-Mercaptoethanol (M3148); ascorbic acid (A5960); N6,2′-O-dibutyryladenosine 3′,5′-cyclic monophosphate sodium salt (D0260). Thermo Fisher Scientific, Waltham, USA: B-27 supplement (17504044); GlutaMAX supplement (35050061); 488-labeled secondary antibodies (S11223), 594-labeled secondary antibodies (S11227); Neurobasal-A Medium, minus phenol red (12349015). Biolegend, San Diego, USA: human Brain-derived neurotrophic factor (788904). Alomone labs, Israel: anti-ryanodine receptor 2 antibody (ARR-002). Abcam, UK: anti-Presenilin 1 antibody (ab15458). Wako Pure Chemical Industries, Osaka, Japan: paraformaldehyde (160–16061); Polyoxyethylene (10) Octylphenyl Ether (160–24751). NACALAI TESQUE, Kyoto, Japan: penicillin and streptomycin (26253–84), VectorLaboratories, Burlingame, USA: normal goat serum (S-1000).

### Preparation of iPSC- derived neural stem cells (NSCs)

Four iPSC-derived NSCs were purchased from Axol Bioscience (Cambridge, UK). Information about the donors was described in the texts and is also available online (https://www.axolbio.com/). Briefly, $PSEN1_{A246E}$ cell line was established from a 31-year-old female bearing *PSEN1* p.A246E with early onset AD (onset age at 45 years-old; AX0114) [17]. $PSEN1_{L286V}$ cell line was established from a 38-year-old male bearing *PSEN1* p.L286V with early onset AD (AX0112). $PSEN1_{M146L}$ cell line was established from a 53-year-old male bearing *PSEN1* p.M146L with AD (AX0113). As a control (WT), a cell line established from a 64-year-old healthy female subject (AX0019) was used. All cells were differentiated into mature neurons for 27 ± 2 days.

Axol Bioscience performed karyotyping of these cells before and after differentiation, and no differences were found. All reagents used for culturing were purchased from Axol Bioscience (Cambridge, UK). The expansion of neural stem cells and neural cell differentiation was performed according to the protocol published by Axol Bioscience (Human iPSC-Derived Neural Stem Cells System A, B, C and D Protocol version 5.0: https://www.cosmobio.co.jp/product/uploads/document/AXO_Protocol_Human_iPSC_Derived_Neural_Stem_Cells.pdf

and Enriched Cerebral Cortical Neurons Derivation from Axol iPSC Neural Stem Cells User Guide V1.4: https://axolbio.com/publications/enriched-cerebral-cortical-neuron-protocol/). Briefly, a 6-well plate was coated with SureBond-XF for at least 4 hours for expansion cultures. Human iPSC-derived NSCs were seeded and expanded in the 6-well plate. The medium was replaced with Neural Maintenance Medium (containing the Neural Maintenance Medium Supplement) adding human FGF2 (a final concentration of 20 ng/mL) and human EGF (a final concentration of 20 ng/mL). Polymer coverslip-bottom dishes were coated with polyornithine aqueous solution (0.05 mg/mL) and SureBond-XF for neuronal differentiation cultures. Human iPSC-derived NSCs were seeded in the polymer coverslip-bottom dishes. From day 1 to day 6 of the culture, the medium was replaced with Neural Maintenance Medium containing NeurOne Supplement A every other day. In addition, Neural Maturation Basal Medium was prepared by adding 1 mL of B-27 supplement, 0.5 mL of GlutaMAX supplement, 25 μL of 2-Mercaptoethanol, and 0.5 mL penicillin and streptomycin (a final concentration of 1%) to 50 mL of Neurobasal-A Medium, minus phenol red. From day 7 to day 14 of the culture, the medium was replaced with Neural Maturation Basal Medium containing NeurOne Supplement B, human Brain-derived neurotrophic factor (BDNF) (a final concentration of 0.02 μg/mL), N6,2′-O-dibutyryladenosine 3′,5′-cyclic monophosphate sodium salt (a final concentration of 0.5 mmol/L), and ascorbic acid (a final concentration of 0.2 mmol/L) every other day. On the 15th day, the whole volume of the medium was replaced with Neural Maturation Basal Medium supplemented with BDNF (a final concentration of 0.02 μg/mL), N6,2'-O-dibutyryladenosine 3′,5′-cyclic monophosphate sodium salt (a final concentration of 0.5 mmol/L), and ascorbic acid (a final concentration of 0.2 mmol/L). After that, half of the medium was replaced every other day.

## Ca$^{2+}$ imaging

The intracellular Ca$^{2+}$ levels were measured using Cal-520 (21130, AAT Bioquest, Inc., Sunnyvale, USA) as previously described [18]. Briefly, iPSC-derived neurons on polymer coverslip-bottom dishes as mentioned in section "Preparation of iPSC-derived neural stem cells (NSCs)" were incubated with the Cal-520/AM (4 μM) at 37˚C with 5% CO$_2$ for 30 min in Neural Maturation Medium. The polymer coverslip-bottom dishes were washed with a Tyrode buffer (in mM:140 NaCl, 5.4 KCl, 5 HEPES, 1.2 MgCl$_2$, 1.8 CaCl$_2$, 10 glucose, pH 7.4). Data acquisition and analysis were performed using AquaCosmos 2.0 (Hamamatsu Photonics, Hamamatsu, Japan). Ethylene glycol-bis($\beta$-aminoethyl ether)-N,N,N',N'-tetra acetic acid (EGTA) (15214–34) was purchased from NACALAI TESQUE (Kyoto, Japan). Cyclopiazonic acid (030–17171), dantrolene (359–44501) and xestospongin c (244–00721) were purchased from FUJIFILM Wako Chemicals (Osaka, Japan). Carvedilol (C184625) was purchased from Toronto Research Chemicals (Toronto, Canada). VK-II-86 (AOB4007) was purchased from AOBIOUS INC (Gloucester, US).

## Western blotting

Proteins were extracted from iPSC-derived neurons with lysis buffer (#9803S, Cell Signaling Technology, Danvers, MA, US), and adjusted to 13 μg in each lane. Electrophoresis was conducted on 7% agarose gels (#4561026Bio-Rad, Hercules, CA, US). Recombinant RyR2 (PrEST Antigen RYR2, APREST73317, Sigma-Aldrich, St. Louis, MO, US) was used as a positive control. Recombinant RyR2 (PrEST Antigen RYR2, APREST73317, Sigma-Aldrich) was used as a positive control. The positive control was diluted to 1180 ng/15 μL per well with 12 μL phosphate buffered saline (PBS, 14190–136, Gibco, Thermo Fisher Scientific, Waltham, MA, US) and 3 μL of loading buffer (NP0008, Invitrogen, Thermo Fisher Scientific). Proteins resolved on gels were electrophoretically transferred to polyvinylidene difluoride (PVDF) membranes at 5

V for 1 hour and 45 minutes. The PVDF membrane was subject to blocking against nonspecific reactions by immersing into a 1% skim milk solution in Tris-buffered saline (#T9141, Takara Bio, Shiga, Japan) with 0.1%Tween 20 (#9005-64-5, TOKYO CHEMICAL INDUSTRY, Tokyo, Japan) for 1 hour at room temperature. Subsequently, it underwent overnight incubation at room temperature with RyR2 antibodies, which was previously synthesized by Murayama et al. [19] and diluted in TBST (1:1000) for enhanced specificity in target protein recognition. The membranes were washed with TBST and incubated with peroxidase-conjugated goat anti-rabbit IgG (Bio-Rad, Hercules, CA, US) at room temperature for 1 hour. After washing membranes, positive bands were detected by Clarity Wester ECL Substrate (#1070501, Bio-Rad).

### RNA-Seq

RNA-Seq was performed as described anywhere [20]. Briefly, differentiated neurons at 28 days were rinsed with cold PBS and scraped using the lysis buffer (LRT) containing β-mercaptoethanol (#135–07522, Wako Chemicals, Osaka, Japan) on ice. RNA extraction was done using RNeasy Plus Mini Kit (Qiagen, 74134). Genomic DNAs were removed using gDNA Eliminator spin columns (Catalog #74134, Qiagen, Hilden, Germany). cDNAs were prepared with Clontech Smart-seq v4 ultra-low kit (Cat# 634888, Takara Bio, Shiga, Japan). PolyA tail enrichment was done using NEBNext Poly(A) mRNA Magnetic Isolation Module (Cat No. E7490). Library preparation and sequencing were performed at Rhelixa using NEBNext UltraII Directional RNA Library Prep Kit (Cat No. E7760). Approximately 15–20 M reads per library were generated using the Illumina NovaSeq 6000 sequencer. The sequenced data were mapped to GRCh38 genome (gencode40) using STAR. Uniquely mapped sequencing reads were assigned to GRCh38 gencode40 genes using featureCounts, and counts data were analyzed using edgeR.

### Statistics

Ca$^{2+}$ signal data were classified into five categories (no signals, narrow wave, wide wave, oscillation, and burst) and their composition ratios were shown using GraphPad Prism software (GraphPad Software, San Diego, USA). Multiple comparisons of population proportions for each category were performed by Tukey's test Comparisons of the rates of Ca$^{2+}$-oscillations and Ca$^{2+}$-bursts in neurons bearing *PSEN1* variants were performed, and their absolute mean differences and 95% and 99% confidence intervals were calculated for each comparison. The statistical significance was defined as $p < 0.05$. The other statistical analyses were performed using EXCEL Toukei Ver.7.0 (ESUMI Co., Ltd, Tokyo, Japan) and EZR (Saitama Medical Center, Jichi Medical University, Saitama, Japan) [21].

## Results

### Differentiation of the iPSCs into neural cells

iPSC-derived neural stem cells (NSCs) from a healthy individual and early-onset AD patients bearing *PSEN1* variants were differentiated into neural cells as described in Methods. Differentiation of neural cells was confirmed by the expression of neuronal markers βIII-tubulin and microtubule associated protein 2 (MAP2) (**S1 Fig**).

### The iPSC-derived neurons bearing *PSEN1* variants exhibited abnormal Ca$^{2+}$-waves

We found that all three types of neurons bearing *PSEN1* variants (p.A246E, p.M146L, and p. L286V) showed abnormal Ca$^{2+}$-waves, most notably Ca$^{2+}$-oscillations and bursts. In contrast, WT neurons showed occasional spontaneous Ca$^{2+}$-waves (**Fig 1A–1G**). **Fig 2** summarizes the

frequency of different Ca$^{2+}$-wave patterns observed in each cell type. The abnormal Ca$^{2+}$-oscillations were more frequently observed in the neurons bearing *PSEN1* variants (p.A246E, p. M146L, and p.L286V) compared to the WT neurons. Furthermore, Ca$^{2+}$-bursts were significantly higher in neurons bearing *PSEN1* variants (p.A246E and p.M146L) compared to the WT neurons, suggesting that these two variants may cause more severe phenotypes than p. L286V since Ca$^{2+}$-bursts are more disordered than Ca$^{2+}$-oscillations.

## Involvement of endoplasmic reticulum (ER) in the abnormal Ca$^{2+}$-bursts

In human neurons, intracellular Ca$^{2+}$ homeostasis is regulated by Ca$^{2+}$ influx via Ca$^{2+}$ channels located in the plasma membranes and Ca$^{2+}$-induced Ca$^{2+}$ release (CICR) from the ER via RYR and inositol 1,4,5-trisphosphate (IP$_3$) receptors [22]. In addition, it has been proposed that PSEN1 variants may form pores in the ER membrane [23], leading to Ca$^{2+}$ leaks into the cytoplasm through these pores [5]. To examine how the variant PSEN1 p.A246E causes Ca$^{2+}$ leaks, the cells were perfused with Ca$^{2+}$-free extracellular solutions containing 10 mM EGTA. Removal of extracellular Ca$^{2+}$ eliminated the abnormal Ca$^{2+}$-bursts (**Fig 3A**). This indicates that abnormal Ca$^{2+}$-waves associated with the PSEN1 p.A246E are CICR-dependent.

To examine whether the ER is involved, cyclopiazonic acid (20 μM), an inhibitor of sarco/endoplasmic reticulum Ca$^{2+}$-ATPase (SERCA), was applied to the Ca$^{2+}$-waves [24], and it had reversibly suppressed the abnormal Ca$^{2+}$-transients in the cells. Since cyclopiazonic acid depletes the Ca$^{2+}$ storage in the ER, these results indicate that the ER is indeed the main resource of the abnormal Ca$^{2+}$ bursts (**Fig 3B**).

## Ryanodine receptor (RyR) 2 facilitates the abnormal Ca$^{2+}$-bursts

It has been reported that inositol triphosphate receptor (IP$_3$R) is a mediator to release Ca$^{2+}$ from the ER [12]. To examine the involvement of IP$_3$R in the abnormal Ca$^{2+}$-bursts [22],

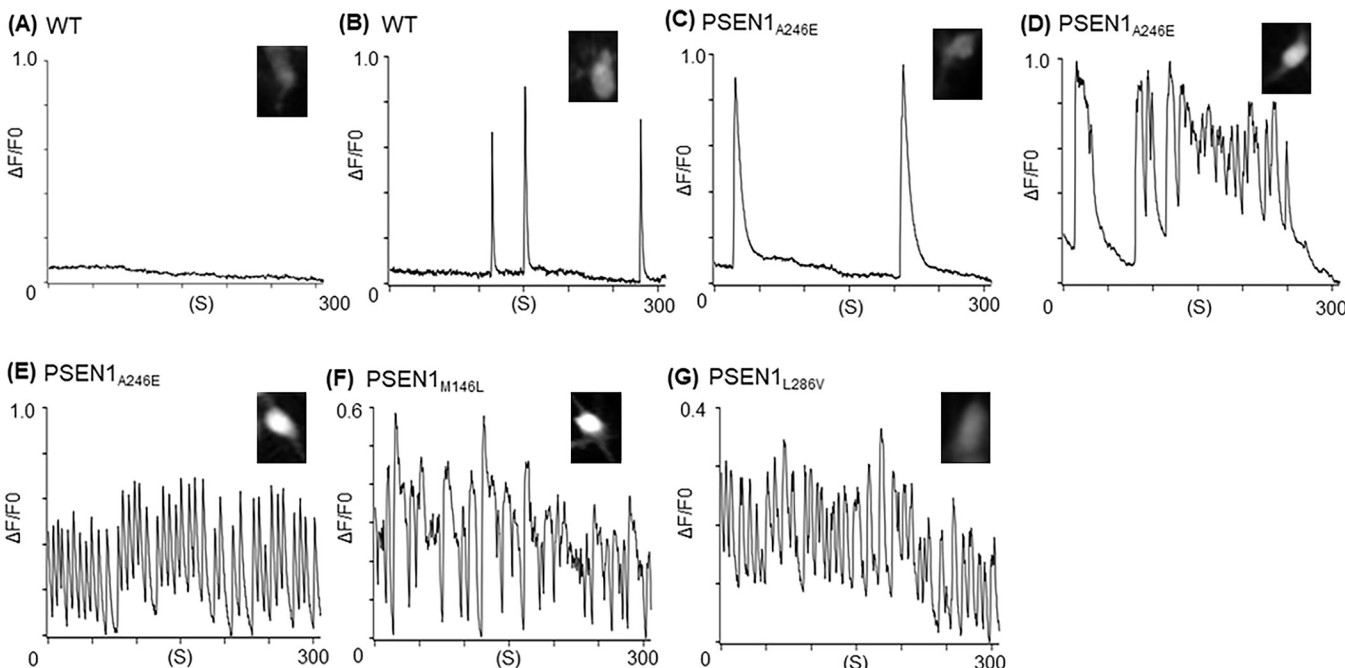

**Fig 1. Representative traces of Ca$^{2+}$ transients and their frequencies observed in various neuronal cells.** (**A**) No Ca$^{2+}$ transients observed in a WT neuron. (**B**) Narrow Ca$^{2+}$ transients observed in a WT neuron. (**C**) Wide Ca$^{2+}$ transients observed in a PSEN1$_{A246E}$ neuron. (**D**) Ca$^{2+}$-oscillations observed in a PSEN1$_{A246E}$ neuron. (**E**) Ca$^{2+}$-bursts observed in a PSEN1$_{A246E}$ neuron. (**F**) Ca$^{2+}$-bursts observed in a PSEN1$_{M146L}$ neuron. (**G**) Ca$^{2+}$-bursts observed in a PSEN1$_{L286V}$ neuron. The fluorescence images stained with Cal-520 for each type of cell were shown in the inset.

xestospongin c (3 μM), an IP₃R antagonist, was applied to the Ca²⁺-bursts in PSEN1₍A246E₎ cells, which did not eliminate the abnormal Ca²⁺-bursts (**Fig 4A**).

Human neurons express RyR1, RyR2, and RyR3 that pump out Ca²⁺ into the cytoplasm from the ER with a predominance of RyR2 [25]. We examined effects of dantrolene, an inhibitor of RyR1 and RyR3 [26], and carvedilol, an inhibitor of RyR2 [27], on the Ca²⁺-bursts in randomly chosen neurons bearing *PSEN1* variants. We found that carvedilol (30 μM), but not dantrolene (30 μM), eliminated the abnormal Ca²⁺-transients in the neuronal cells bearing *PSEN1* p.A246E (**Fig 4B and 4C**). The involvement of RyR2 was also confirmed with VK-II-86 (10 μM), a RyR2 inhibitor without the β-blocking effects (**Fig 4D**) [27]. The Ca²⁺-bursts were also eliminated by carvedilol in the PSEN1₍M146L₎ and PSEN1₍L286V₎ cells (**S2 Fig**). The responses

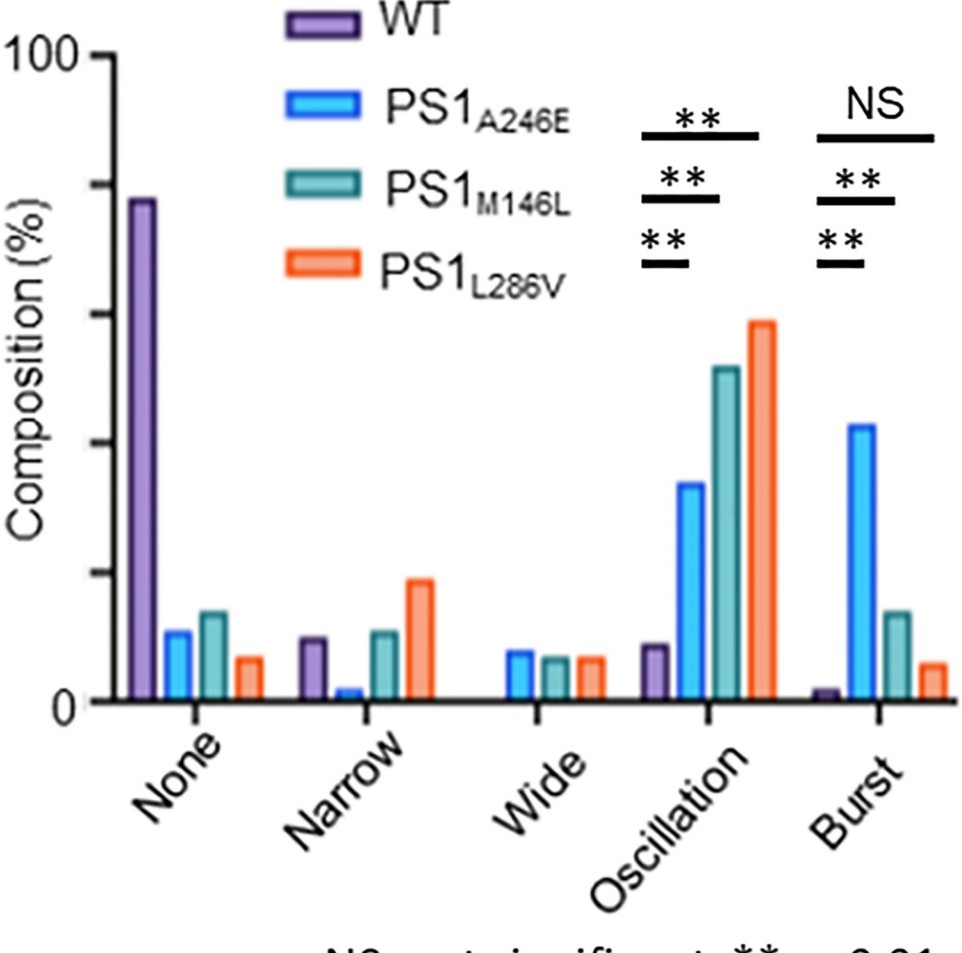

**Fig 2. The frequencies of different types of Ca²⁺ transient in WT and *PSEN1*-mutated (p.A246E, p.M146L, p. L286V) neuronal cells.** Numbers of cell counts for WT and *PSEN1*-mutated neurons were as follows: WT, 592; PSEN1₍A246E₎, 353; PSEN1₍M146L₎, 155; and PSEN1₍L286V₎, 363. Ca²⁺ transients were recorded at a sampling rate of 370 ms. Comparisons in the rates of Ca²⁺-oscillations and Ca²⁺-bursts in neurons bearing PSEN1 variants were performed between the p.A246E and WT neurons, between the p.M146L and WT neurons, and between the p.L286V and WT neurons. As for the Ca²⁺-oscillations, the 99% confidence intervals (CI) of these comparisons were all significant (**S1 Table**). Comparisons in the rates of Ca²⁺-bursts were [99%CI: -0.483, -0.338] (p.A246E and WT) and [99%CI: -0.224, -0.020] (p.M146L and WT), respectively, both of which were significant (*p*<0.01). Meanwhile, a comparison in the rates of Ca²⁺-bursts in neurons with PSEN1 variants between the p.L286V and WT neurons showed [95%CI: -0.100, 0.019], which was nonsignificant (**S2 Table**). NS depicts non-significant; **$p$<0.01.

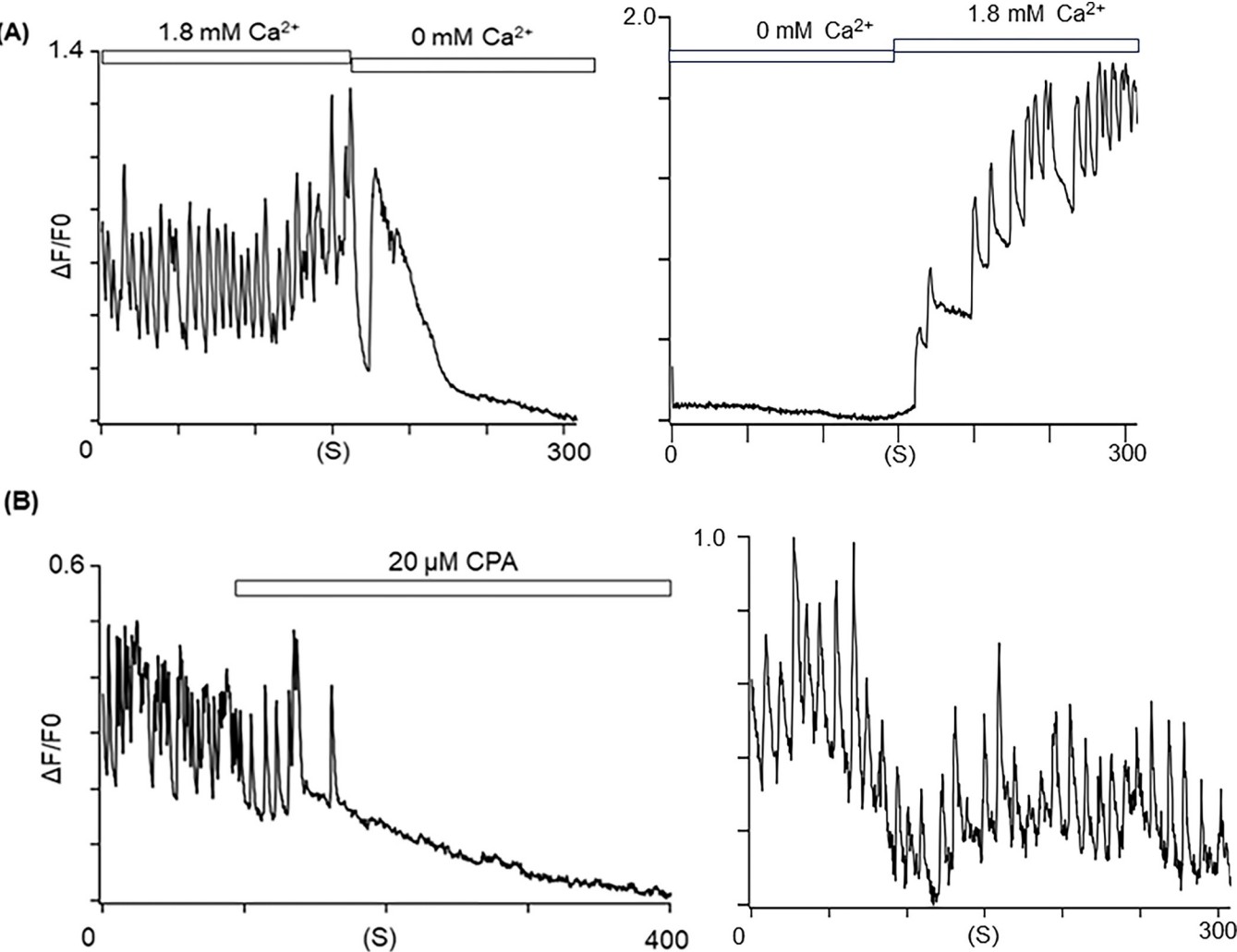

**Fig 3. Involvement of endoplasmic reticulum in the underlying mechanisms of Ca$^{2+}$ bursts in the PSEN1$_{A246E}$ neurons.** (**A**) Ca$^{2+}$-free extracellular solutions containing EGTA eliminated the Ca$^{2+}$-bursts in the PSEN1$_{A246E}$ neuron (left panel). A trace after washout (right panel). Note that these are not continuous recordings but the trace in the right panel was obtained within several minutes after washout in the same cell. (**B**) Application of 20 μM CPA eliminated the Ca$^{2+}$-bursts in the PSEN1$_{A246E}$ neuron (N = 27). A trace after washout (right panel).

to each reagent were all-or-none. These results indicate that RyR2 is the main facilitator of the abnormal Ca$^{2+}$-bursts in the cells bearing *PSEN1* variants.

## The expression levels of RyR2 were not affected by *PSEN1* p.A246E

In a presenilin-knock out mouse, the protein level of RyR was reduced, which affected the intracellular Ca-homeostasis [28]. It has also been reported that PSEN1 physically interacts with RyR2 [29]. To examine the effects of the PSEN1 variant on RyR2 levels, we quantified the levels of PSEN1 and RyR2 using Western blot. Our data revealed that the expression levels of RyR2 and PSEN1 were similar both in the WT and PSEN1$_{A246E}$ cells (**Fig 5**).

## RNA-Seq

In addition, the gene expressions were measured by RNA-Seq, and the results were compared between the WT and PSEN1$_{A246E}$ neurons (n = 3 for each cell type). Genes with FDR < 0.05

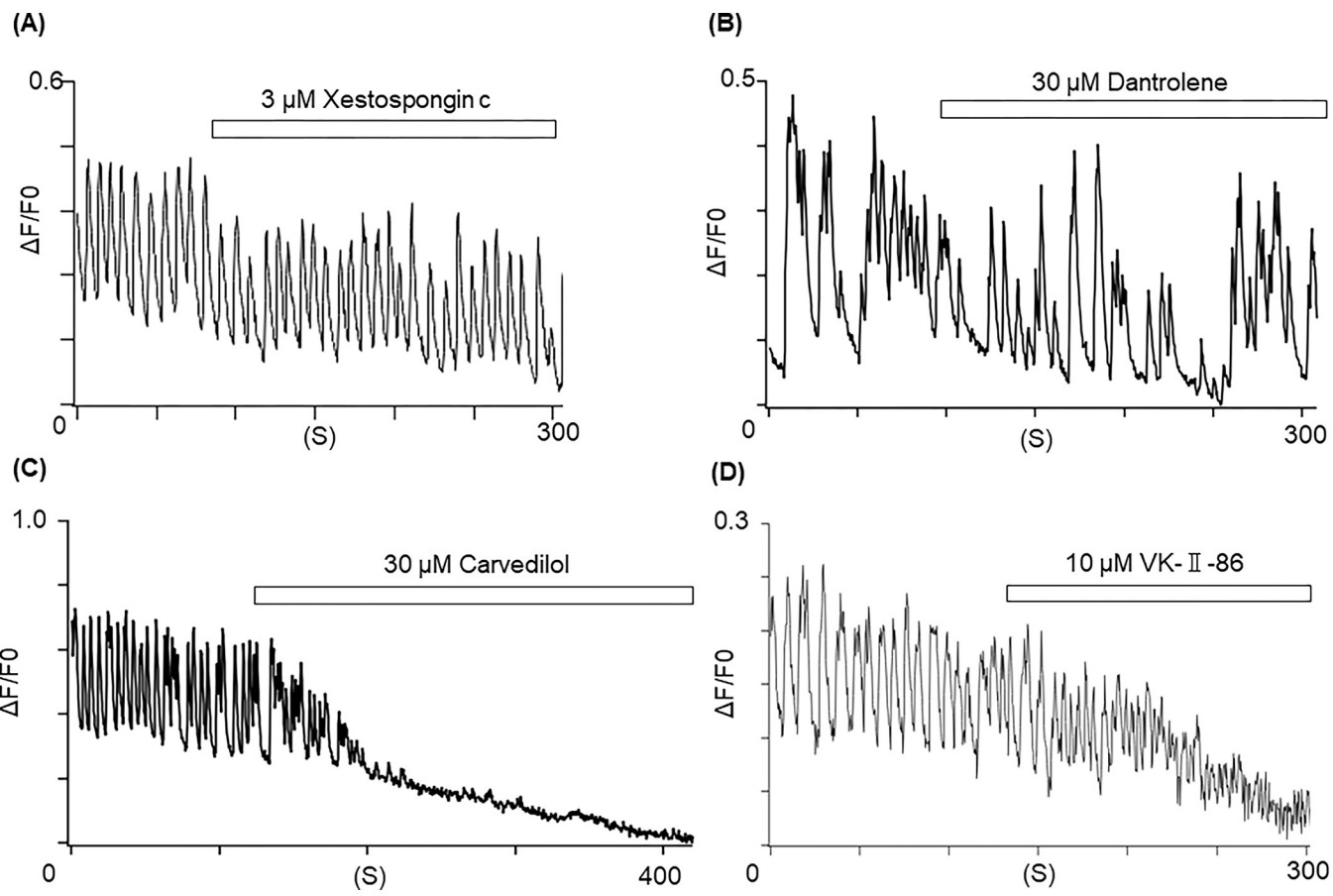

**Fig 4. RyR2 facilitates the Ca²⁺ bursts from the ER in the PSEN1$_{A246E}$ neurons.** (**A**) Xestospongin c (3 μM) did not affect the Ca²⁺-bursts in the PSEN1$_{A246E}$ neuron (N = 20). (**B**) Dantrolene (30 μM) did not affect the Ca²⁺-bursts in the PSEN1$_{A246E}$ neuron (N = 21). (**C**) Carvedilol (30 μM) terminated the Ca²⁺-bursts in the PSEN1$_{A246E}$ neuron (N = 98). (**D**) VK-II-86 (10 μM) terminated the Ca²⁺-bursts in the PSEN1$_{A246E}$ neuron (N = 18).

were selected, recovering a list of the top 10000 differentially expressed genes (DEGs) with 1506 upregulated and 1521 downregulated in the PSEN1$_{A246E}$ cells (**S3 Table,** FDR < 0.05).

**Fig 6** shows a smear plot of the alteration of the gene expression, annotated for genes associated with ER and mitochondrial-stress and glycolysis [30]. Among the top 10000 DEGs, the *MCUB* gene encoding for the mitochondrial calcium uniporter dominant negative subunit β was upregulated, while the gene *MTLN* encoding mitoregulin and the gene *BCL2A1* encoding BCL2 related protein A1 were downregulated in the PSEN1$_{A246E}$ cells. Also, the *RYR2* and *RYR3* genes encoding RyRs were downregulated in the PSEN1$_{A246E}$ cells, which may reflect negative feedback of Ca²⁺ overload.

## Discussion

Our data showed abnormal Ca²⁺-bursts recorded in human neuronal cells obtained from three-unrelated early-onset AD patients bearing *PSEN1* variants (p.A246E, p.M146L, or p. L286V). We also demonstrated that the abnormal Ca²⁺-bursts were suppressed by carvedilol, a known RyR2 inhibitor, as well as VK-II-86, a RyR2 inhibitor without the β-blocking effects [27]. Importantly, these *PSEN1* variants do not produce significant amount of abnormal Aβ [9], which suggests that amyloid-independent pathophysiology plays a role in early-onset AD.

## Does neuronal hyperexcitation cause early-onset AD?

Although epileptic activity is often associated with AD, whether it is a cause, or a consequence of other factors, such as abnormal Aβ and Tau proteins, remains unknown [2]. However, seizures and myoclonus occur frequently in early-onset AD patients who have yet to develop severe dementia [1, 31]. This progression may likely be a pathological condition of cardiomyopathy. In cardiomyopathy, it is often seen that abnormal electrical activities (i.e., arrhythmias) precede structural remodeling as how seizures progress to severe dementia [32, 33].

Just as Lam et al. proposed that silent seizure-like signals cause AD, our data strongly promotes a new underlying mechanism of *PSEN1*-associated early-onset AD. Since their patients did not bear *PSEN1* variants, the underlying mechanism of the neuronal hyperexcitation observed is unknown. However, *PSEN1* variants p.A246E and p.L286V tested in this study were found in AD patients who had seizures [34], which is notably consistent with Lam's findings regardless of the causes of neuronal hyperexcitation. In addition, the *PSEN1* p.A246E and p.L286V as well as p.M146L variants did not produce significant amount of abnormal Aβ peptides [9], further supporting the amyloid-independent theory. Thus, it is reasonable to propose that neuronal hyperexcitation associated with Ca²⁺-dysregulation is one of the major amyloid-independent mechanisms of early-onset AD associated with *PSEN1* [3].

## The underlying mechanisms of Ca²⁺ dysregulation in *PSEN1*-related AD

There are numerous studies that Ca²⁺ overload plays an important role in damaging neurons though this topic is still controversial. For instance, Ca²⁺-leak from ER induces cell death by increasing the permeability of mitochondrial inner membranes [35]. Müller et al. proposed that constitutively activated transcription factor cAMP response element binding protein (CREB) associated with activation of IP₃R through CaM kinase β and CaM kinase IV pathways cause cell death and sensitivity to Aβ toxicity [36]. In human iPSC-induced neurons from an AD patient bearing *PSEN1* p.A246E, increased Aβ₄₂ and abnormal Ca²⁺ surge were observed, and these were reversed by dantrolene [15], which is inconsistent with our data (**Fig 4B**). This can be due to differences in creating iPSCs and patient background.

Nonetheless, an important question moving forward is "How can we link local seizures due to abnormal Ca²⁺-bursts with cognitive behavioral disorders?" There are several suggestive reports. In an AD-mouse model (5xFAD) that bear multiple amyloid β precursor gene (*APP*) and *PSEN1* variants, reducing Ca²⁺ release from the ER via replacing the WT RyR2 with a mutated one prevented the hyperactivity of pyramidal neurons [14]. SanMartin et al. reported that amyloid β oligomers (AβOs) amplified RyR-mediated Ca²⁺, which induced mitochondrial dysfunctions in primary hippocampal cultures obtained from 18-day-old Sprague-Dawley rats [13]. Later, the same group found that N-acetylcysteine prevented the AβO-induced mitochondrial dysfunctions and defective structural changes in Sprague-Dawley juvenile male rats [37]. In another study, long-term treatment with carvedilol suppressed seizures in amyloid precursor protein transgenic mice (TgCRND8), a well-established AD model [38]. In contrast, Shilling et al. showed that, in a mouse model bearing *PSEN1* p.M146V, reduction of IP₃R1 expression restored CREB-dependent gene expression rescuing the abnormal hippocampal long-term potentiation. Thereby, they propose that the IP₃R1 is the therapeutic target in *PSEN1*-associated AD [12]. However, in our data, Xestospongin c, an inhibitor of IP₃R, did not suppress the abnormal Ca²⁺-bursts (**Fig 4A**).

Our RNA-Seq data demonstrated that mitoregulin encoded by the *MTLN* that regulates mitochondrial membrane potential [39] and BCL2A1 encoded by the *BCL2A1* that is

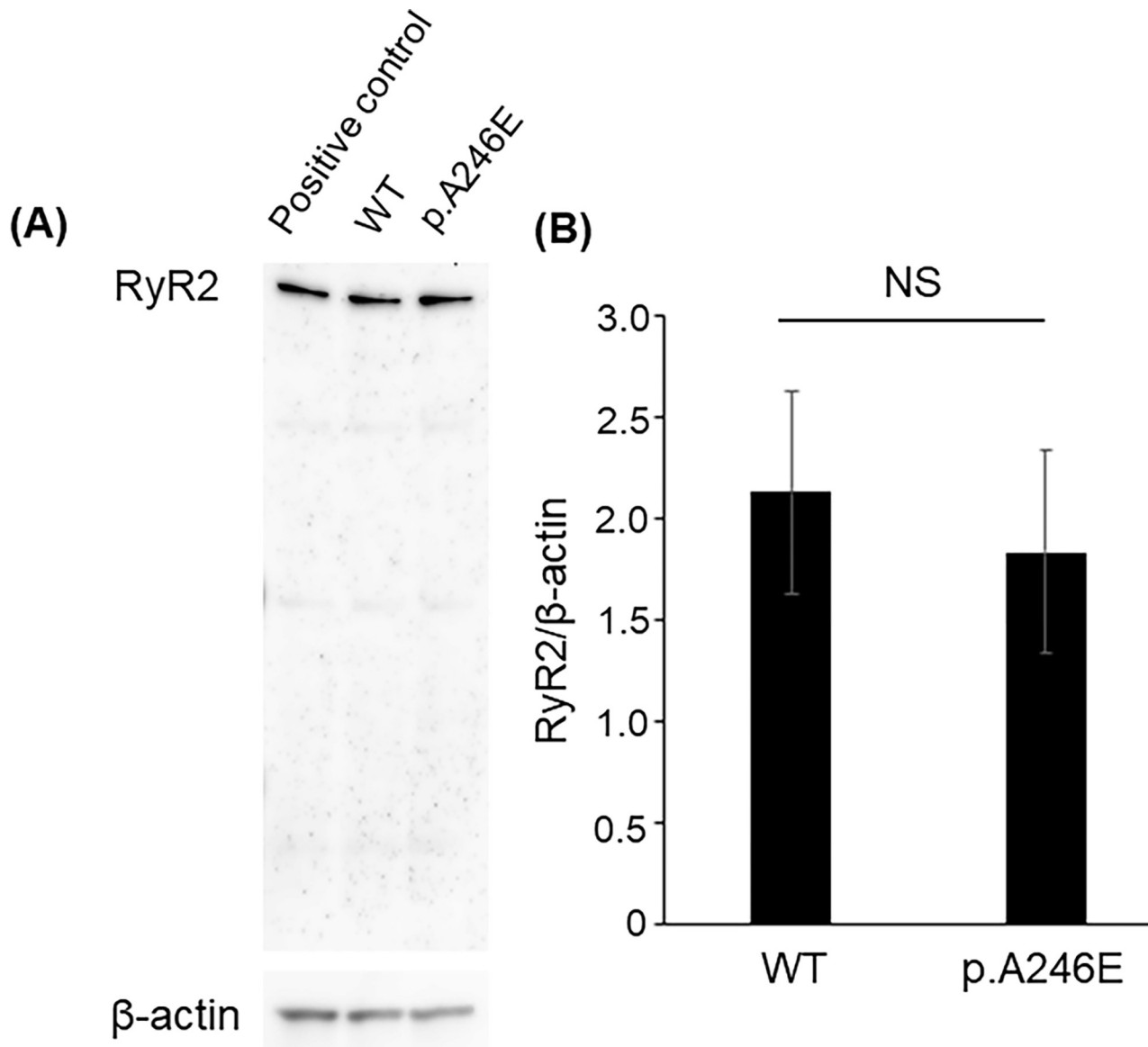

**Fig 5. Western blotting to measure RyR2 levels in WT and PSEN1$_{A246E}$ neural cells.** (**A**) A representative staining. Each lane was loaded with positive control (left), WT sample (mid), and PSEN1$_{A246E}$ (right) sample, respectively. The upper bands show staining with Anti-RyR2 antibody, and the lower bands show staining with Anti-β-actin antibody. (**B**) A bar graph shows the expression of RyR2 in the WT and PSEN1$_{A246E}$ neurons. The experiments were repeated three times. The statistical analysis was performed by Mann-Whitney U test.

associated with apoptosis in brains [40] were significantly downregulated in the PSEN1$_{A246E}$ neurons. In addition, ER stress pathway genes such as *EIF2S1* and *ATF4* [30], and mitochondrial stress pathway genes such as *MCU*, *CASP3*, and *CASP9* tended to be upregulated in the PSEN1$_{A246E}$ compared to the WT neurons. In contrast, the *VDAC1* encoding the voltage-dependent anion channel 1 (VDAC1) in the outer mitochondrial membrane tended to be downregulated in the PSEN1$_{A246E}$ [41]. Although these changes are not statistically significant (FDR > 0.05), it suggests that Ca²⁺-overload due to the abnormal Ca²⁺-bursts may damages cytosolic organelles, leading to neural malfunctions. Indeed, *RYR2* as well as *RYR3* were down-regulated, which might have occurred by unknown negative feedback pathways to reduce

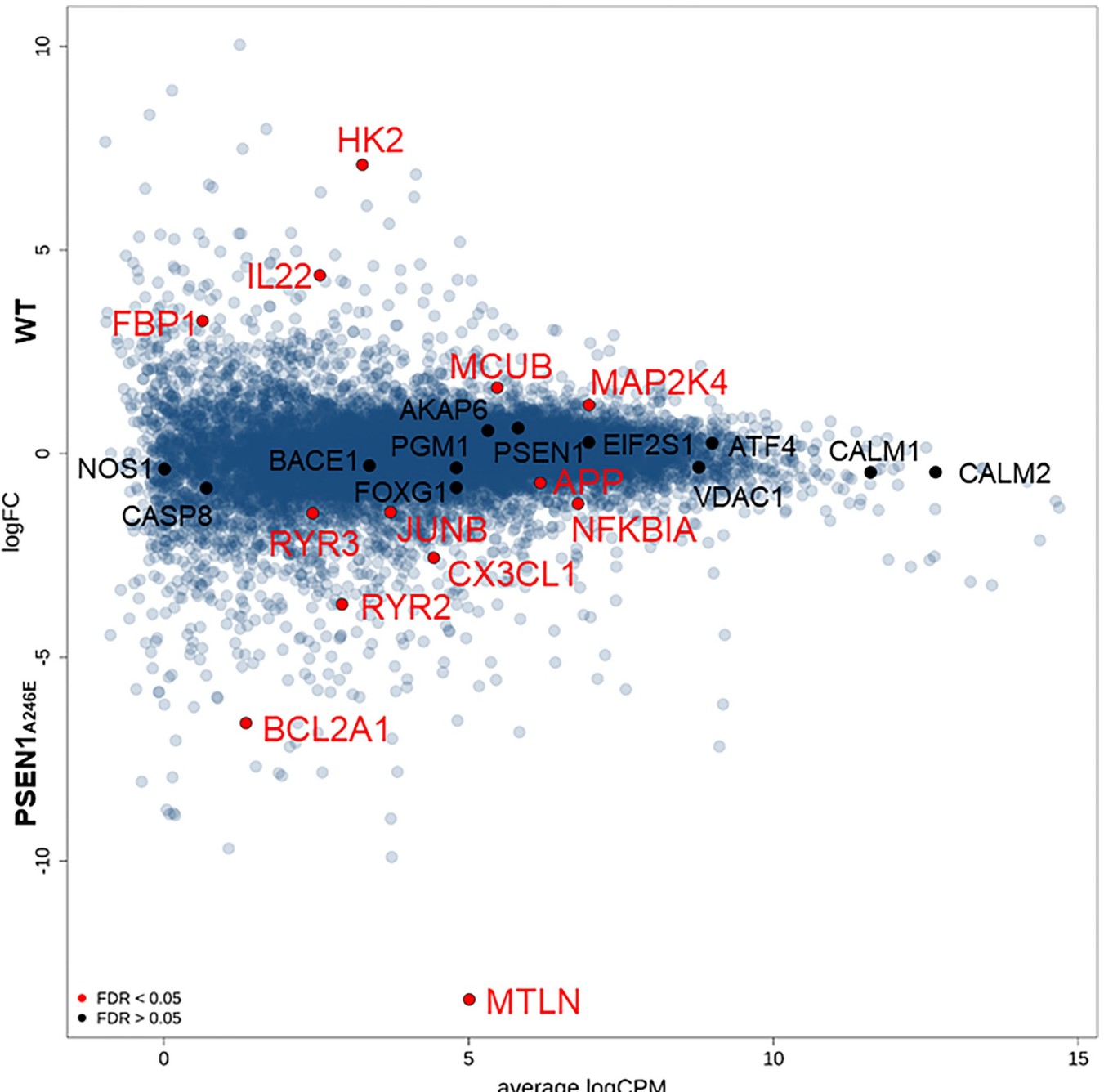

**Fig 6. Comparisons of gene expressions between PSEN1$_{A246E}$ and WT neurons evaluated by RNA-Seq.** A smear plot of detected gene expressions. Relative gene expression levels by comparisons of the PSEN1$_{A246E}$ cells against the WT neurons are plotted. Several genes of interest were indicated by solid red symbols (FDR < 0.05).

intracellular and mitochondrial Ca²⁺ overload. Thus, currently, we are investigating the functional changes in the ER and mitochondria in human iPSC-induced neurons.

Though actual pathogenic mechanisms of AD are still controversial, it is reasonable to theorize that abnormal Ca²⁺-bursts through RyR2 can be one of the main pathophysiological mechanisms in early-onset AD associated with *PSEN1*.

## Limitations

There are several significant limitations: (1) we used human iPSC-induced neural cells whose anatomical and functional characteristics can be different from actual human neurons; (2) it is difficult to generate homogeneous neural cell populations though we used a protocol to differentiate the cells into cerebral cortical neural cells; (3) although we examined the neural cells bearing three different *PSEN1* variants, we do not know if our hypothesis can be applied to AD caused by other factors such as *PSEN2* and *APP* variants; (4) we do not know the effects of these *PSEN1* variants on gating-kinetics of RyR2 due to lack of experiments in human brain cells. Also, we do not know the underlying mechanisms by which dantrolene did not suppress RyR2 in our system [42]. However, those recent data were mainly obtained in cardiomyocytes, not in human neurons [42–44]. At this moment, further investigation is beyond the scope of this study; (5) our data still did not reveal how abnormal Ca$^{2+}$-bursts cause early-onset AD (i.e., neural damages). Currently, we are studying the effects of Ca$^{2+}$-bursts on ER and mitochondrial functions; (6) the RNA-Seq data showed that various pathways were affected by *PSEN1* p.A246E (listed in **S1 Table**). Currently, we do not know the association of these changes and AD phenotypes. We will further investigate the roles of the affected various pathways in pathophysiological mechanisms of AD; and (7) a clinical trial failed to show the effects of carvedilol in AD (https://www.clinicaltrials.gov/study/NCT01354444?tab=history). Probably this was due to small population (n = 14), and we expect that carvedilol may not be able to reverse already damaged neurons. Interestingly, a recent large cohort study in Denmark indicated that the prevalence of AD was significantly lower in the patients who took carvedilol than the patients who took atenolol and bisoprolol [45]. However, it was unclear if the pathophysiology in these patients was uniform.

## Conclusions

Despite these limitations, we propose that abnormal Ca$^{2+}$ signals play a role in the pathophysiological mechanisms in early-onset AD associated with *PSEN1*. Our data also elucidated that RyR2 can be a potential therapeutic target in prevention of AD. Further large-scale randomized clinical studies are warranted to validate our proposals.

## Supporting information

**S1 Fig. Immunocytochemical staining of WT and PSEN1 p.A246E neurons.** Immunocytochemical staining of WT and PSEN1 p.A246E neurons. Images obtained from WT neurons (A) and PSEN1 p.A246E neurons (B). The far-left panels are negative controls without using primary antibodies. Blue, DAPI; Red, β-tubulin III; Green, MAP2; Scale bar, 10 μm.
(DOCX)

**S2 Fig. Effects of carvedilol on the Ca$^{2+}$-bursts in the PSEN1$_{M146L}$ and PSEN1$_{L286V}$ neurons.**
(DOCX)

**S1 Table. Statistics for Ca$^{2+}$-oscillation in Fig 2.** NS: not significant, **p$<$0.01, CI: confidence interval, LL: lower limit, UL: upper limit.
(DOCX)

**S2 Table. Statistics for Ca$^{2+}$-bursts in Fig 2.** NS: not significant, **p$<$0.01, CI: confidence interval, LL: lower limit, UL: upper limit.
(DOCX)

**S3 Table. A list of the top 10,000 genes that were affected in the PSEN1$_{A246E}$ neurons compared to the WT neuron.**
(DOCX)

## Acknowledgments

We are thankful to the Laboratory of Molecular and Biochemical Research, Biomedical Research Core Facilities and the Laboratory of Morphology and Image Analysis, Research Support Center, Juntendo University Graduate School of Medicine, for technical assistance.

## Author Contributions

**Conceptualization:** Takashi Murayama, Tomohiko Ai.

**Data curation:** Atsushi Hori, Haruka Inaba, Takashi Hato, Kimie Tanaka, Shoichi Sato, Mizuho Okamoto, Yuna Horiuchi, Faith Jessica Paran, Corina Rosales.

**Formal analysis:** Atsushi Hori, Takashi Hato, Kimie Tanaka, Shoichi Sato, Mizuho Okamoto, Faith Jessica Paran, Shusuke Mori.

**Funding acquisition:** Atsushi Hori, Tomohiko Ai.

**Investigation:** Atsushi Hori, Takashi Hato, Yuna Horiuchi, Tomohiko Ai.

**Methodology:** Atsushi Hori, Haruka Inaba, Takashi Hato, Kimie Tanaka, Shoichi Sato, Yuna Horiuchi, Faith Jessica Paran, Wado Akamatsu, Takashi Murayama, Nagomi Kurebayashi.

**Project administration:** Tomohiko Ai.

**Software:** Takashi Hato, Shoichi Sato.

**Supervision:** Yoko Tabe, Wado Akamatsu, Takashi Sakurai, Tomohiko Ai, Takashi Miida.

**Visualization:** Yoko Tabe, Takashi Miida.

**Writing – original draft:** Takashi Hato, Mizuho Okamoto, Tomohiko Ai.

**Writing – review & editing:** Takashi Hato, Shusuke Mori, Corina Rosales, Tomohiko Ai.

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
