## [Decision Letter · Decision Letter 0]

9 Oct 2023

PONE-D-23-23711Carvedilol suppresses ryanodine receptor-dependent Ca2+ bursts in human neurons bearing PSEN1 variants found in early onset Alzheimer’s diseasePLOS ONE

Dear Dr. Ai,

Thank you for submitting your manuscript to PLOS ONE. After careful consideration, we feel that it has merit but does not fully meet PLOS ONE’s publication criteria as it currently stands. Therefore, we invite you to submit a revised version of the manuscript that addresses the points raised during the review process.

We look forward to receiving your revised manuscript.

Kind regards,

Pan Li, PhD

Academic Editor

PLOS ONE

Journal Requirements:

"Japan Society for the Promotion of Science (JSPS) grant 20K16532 (AH)

JSPS grant 18K08900 and 21K08135 (TA)"

Reviewers' comments:

Reviewer's Responses to Questions

**Comments to the Author**

1. Is the manuscript technically sound, and do the data support the conclusions?

Reviewer #1: Partly

Reviewer #2: Partly

2. Has the statistical analysis been performed appropriately and rigorously? 

Reviewer #1: Yes

Reviewer #2: I Don't Know

3. Have the authors made all data underlying the findings in their manuscript fully available?

Reviewer #1: Yes

Reviewer #2: Yes

4. Is the manuscript presented in an intelligible fashion and written in standard English?

Reviewer #1: Yes

Reviewer #2: No

5. Review Comments to the Author

Reviewer #1: The manuscript of Hori et al. entitled:”Carvedilol suppresses ryanodine receptor-dependent Ca2+ bursts in human neurons bearing PSEN1 variants found in early onset Alzheimer’s disease” attempts to support the importance of amyloid β (Aβ)-independent mechanisms in the pathogenesis of early-onset Alzheimer’s disease (AD). The authors focused on a potential role of Ca2+ dysregulation in neuronal cells derived from early-onset AD patients harboring three variants of Presenilin-1 (A246E, L286V, and M146L). I agree that much attention has been given to the Aβ hypothesis of AD, however, an increasing body of evidence suggests that dysregulation of neuronal Ca2+ homeostasis may play a fundamental role in the pathogenesis of AD, at least in a sporadic form. Thus, it is feasible to investigate whether neuronal Ca2+ dysregulation might be implicated also in early-onset AD. Reading the manuscript, I found some major and minor issues that need to be addressed by the authors.

MAJOR issues

1. Fig. 4: I was quite surprised that dantrolene did not have any effect on Ca2+ oscillations. For a long time, dantrolene has been consider a selective inhibitor of the RyR1 and RyR3 channels. However, it seems that dantrolene is able to inhibit also the RyR2 channel. This interaction is very likely conditioned by phosphorylation of the channel and/or the presence of associated protein, FKBP12 (DOI: 10.1016/j.jjcc.2019.08.020; 10.1016/s1388-9842(99)00017-3; 10.1016/j.hroo.2020.03.004; 10.1371/journal.pone.0125366; 10.1085/jgp.202213277). And indeed, increased phosphorylation of the RyR2 channel in the AD brains has been reported (DOI:10.1007/s00401-017-1733-7). Thus, I would expect a strong effect of dantrolene. This issue has to be addressed by the authors and mentioned works should be cited.

2. Fig.5: To assess whether protein amounts for the RyR2 channel and PSEN1 were changed in PSEN1-A246E cells comparison to WT control is more appropriate to use standard Western blotting, especially in the case of the RyR2 channel. This is because the anti-RyR antibody used also labels a 130 kDa band, which is not the RyR channel. This non-RyR binding interferes with the RyR-related signal in confocal images. Furthermore, I have doubts about the RyR2-specificity of the antibody, as I did not find satisfactory information on the supplier's website.

3. In connection to Fig.5: The amount of the RyR2 channel in PSEN1-A246E cells versus WT control was not changed when analyzed from confocal images, however, RNA-Seq showed downregulation of RYR2 and RYR3 gens in PSEN1-A246E cells. How could be this finding interpreted?

4. The authors mentioned in the Introduction and Discussion sections that PSEN1 is a Ca2+ pore. However, they did not cite the most relevant papers, which directly showed conductive properties of PSEN1 using the BLM method (DOI: 10.1074/jbc.M111.243063; 10.1016/j.Cell.2006.06). These works should be cited.

MINOR issues

1. Fig. 2: I recommend to show also representative images of Cal-520 filled cells.

2. Fig. 6: The resolution of this figure has to be enhanced because labelling is not legible in the submitted version.

Reviewer #2: This work evaluated the role of ryanodine and IP3 receptors in the generation of calcium signals generated in cell lines derived from patients with familial forms of Alzheimer's disease with a mutation in presenilin 1. Although the results are of interest and can contribute to the discussion involving these receptors in Alzheimer's disease, several issues should be addressed before publication:

1) For example, the authors mention that" human neurons express RyR1 and RyR2" and ignore the RyR3 isoform, which is a brain isoform. They use inhibitors for RyR1 and RyR2 and conclude that RyR2 is very important, but they did not discuss the possible implications of RyR3. Moreover, in the hippocampal neurons, a brain region centrally involved in AD, low levels of RyR1 are present, while RyR3 is highly expressed.

2) They also mentioned a study using dantrolene, an inhibitor of RyR1, as capable of eliminating both Aß42 and Ca2+ signals. However, they dismiss these findings, mentioning that these data still support the classical amyloid theory, which may not be the primary pathological mechanism. It is important to note that although a discussion has been made on the centrality of amyloid to AD pathogenesis, the amyloid cascade is still the leading hypothesis to explain the disease.

3) Still on this matter, the authors omit a critical paper (https://doi.org/10.3389/fnmol.2017.00115) showing that Ca2+ release mediated by the RyR2 isoform causes the deleterious effects of AβOs on mitochondrial function in hippocampal neurons. Knockdown of RyR2 decreased by 40% Ca2+ release induced by the RyR agonist 4-chloro-m-cresol and significantly reduced the cytoplasmic and mitochondrial Ca2+ signals and the mitochondrial fragmentation induced by AβOs. Also, in another work from the same group (10.3389/fnagi.2018.00399), it was shown that AβOs injections directly into the hippocampus decreased RyR2 protein content but increased single RyR2 channel activation by Ca2+ and caused considerable spatial memory deficits. These articles should be cited and discussed correctly, together with Yao'paaper, in the manuscript.

3) Figure 1 is a characterization of the ability to generate neurons from cell lines with previously described protocols, so it should be considered a supplementary figure. The figure legends do not specify the rum or the statistical test used.

4) Regarding Figure 2, the authors mention that Ca2+-bursts were significantly higher in neurons bearing PSEN1 variants (p.A246E and p.M146L) compared to the WT neurons, suggesting that these two variants may cause more severe phenotypes than p.L286V. Although they show a quantification, no statistical analysis was presented to conclude that Ca2+ bursts were higher in those cases than the others.

5) In Figure 3, the results are coherent, but what were the controls to check that BAPTA and CPA did not kill the cells after their appliance to the cultures?

6) In the figure 4, they used carvedilol to block RyR2. Using genetic tools to imply RyR2 in the Ca2+ signals would be relevant. Also, although n is shown for each condition in the figure legend, statistical analysis needs to be mentioned, which is needed to make conclusions about the role of RyR2 as the "main facilitator of the abnormal Ca2+-bursts in the cells bearing PSEN1 variant.". Moreover, significant differences exist among the N used in the different conditions. They mention N = 21 in B, N = 98i C, and N = 18 in D. Given this disparity and the elevated values for N, it is necessary to describe what the authors consider an N.

7) The authors should describe how they performed quantifying the immunofluorescence images in Figure 5.

8) RyR3 was omitted until Figure 6, where it appears to be decreased just as RyR2, but without any comment. Also, the scheme shown in Figure 6B is speculative and can not be supported by the results presented in this work. Moreover, the resolution is inacceptable in this figure.

6. PLOS authors have the option to publish the peer review history of their article (what does this mean?). If published, this will include your full peer review and any attached files.

Reviewer #1: No

Reviewer #2: No

---

## [Author Response · Author response to Decision Letter 0]

28 Mar 2024

Reviewer #1: The manuscript of Hori et al. entitled:”Carvedilol suppresses ryanodine receptor-dependent Ca2+ bursts in human neurons bearing PSEN1 variants found in early onset Alzheimer’s disease” attempts to support the importance of amyloid β (Aβ)-independent mechanisms in the pathogenesis of early-onset Alzheimer’s disease (AD). The authors focused on a potential role of Ca2+ dysregulation in neuronal cells derived from early-onset AD patients harboring three variants of Presenilin-1 (A246E, L286V, and M146L). I agree that much attention has been given to the Aβ hypothesis of AD, however, an increasing body of evidence suggests that dysregulation of neuronal Ca2+ homeostasis may play a fundamental role in the pathogenesis of AD, at least in a sporadic form. Thus, it is feasible to investigate whether neuronal Ca2+ dysregulation might be implicated also in early-onset AD. Reading the manuscript, I found some major and minor issues that need to be addressed by the authors.

Response: We appreciate your time and efforts to review our manuscript and providing us with constructive feedback.

MAJOR issues

1. Fig. 4: I was quite surprised that dantrolene did not have any effect on Ca2+ oscillations. For a long time, dantrolene has been consider a selective inhibitor of the RyR1 and RyR3 channels. However, it seems that dantrolene is able to inhibit also the RyR2 channel. This interaction is very likely conditioned by phosphorylation of the channel and/or the presence of associated protein, FKBP12 (DOI: 10.1016/j.jjcc.2019.08.020; 10.1016/s1388-9842(99)00017-3; 10.1016/j.hroo.2020.03.004; 10.1371/journal.pone.0125366; 10.1085/jgp.202213277). And indeed, increased phosphorylation of the RyR2 channel in the AD brains has been reported (DOI:10.1007/s00401-017-1733-7). Thus, I would expect a strong effect of dantrolene. This issue has to be addressed by the authors and mentioned works should be cited.

Response: We are aware that the effects of dantrolene on the RyR2 channels are controversial. However, the aim of this study is not to investigate the inconsistent results and opinions among the researchers in this field.

The proposal by Kobayashi et al. is a clinical trial to examine whether dantrolene can suppress cardiac arrhythmias [PMID: 31866190]. In another study by Meissner, in human papillary muscles resected from failing hearts dantrolene improved the myocardium-responsiveness to the β-stimulation which did not show direct effects on RyR2 [PMID: 10937928]. Nofi et al. reported that dantrolene attenuated cardiac dysfunction and reduced atrial fibrillation inducibility in a post-myocardial infarction heart failure rat model [PMID: 34113867]. Weel et al. claimed the role of FKBP12.6 in inhibition of RyR2 by dantrolene [PMID: 37279522]. This experiment was done using artificial lipid bilayers, which is a different environment from actual human cells. In addition, the association of FKBP12.6 and RyR2 is controversial [PMID: 17921453]. Furthermore, the role of phosphorylation of RyR2 in heart failure is also controversial [PMID: 24723657; PMID: 22302785].

In contrast, many studies showed that dantrolene does not affect the RyR2 or cardiac functions though these experiments were also performed using animal models or cell lines [e.g., PMID: 966155; PMID: 1015907; PMID: 29674523].

Our data showed the fact that dantrolene did not affect the Ca2+-bursts in human iPSC-induced neurons bearing PSEN1 variants. At this moment, we do not know what the underlying mechanism is. However, even if dantrolene can suppress the Ca2+-bursts by some interventions in our systems, it will not affect our conclusion that carvedilol suppresses the Ca2+-bursts. We added descriptions regarding these issues in the discussion section (Page 20). 

2. Fig.5: To assess whether protein amounts for the RyR2 channel and PSEN1 were changed in PSEN1-A246E cells comparison to WT control is more appropriate to use standard Western blotting, especially in the case of the RyR2 channel. This is because the anti-RyR antibody used also labels a 130 kDa band, which is not the RyR channel. This non-RyR binding interferes with the RyR-related signal in confocal images. Furthermore, I have doubts about the RyR2-specificity of the antibody, as I did not find satisfactory information on the supplier's website.

Response: Fig. 5 was replaced by the Western blot data. We used a different antibody made by our coauthors [PMID: 12805094] in the new experiments. The description was rewritten (Pages 15 and 16).

3. In connection to Fig.5: The amount of the RyR2 channel in PSEN1-A246E cells versus WT control was not changed when analyzed from confocal images, however, RNA-Seq showed downregulation of RYR2 and RYR3 gens in PSEN1-A246E cells. How could be this finding interpreted?

Response: Persistently high intracellular Ca2+ may downregulate the expression of RYR2 and RYR3 although we do not know the exact mechanisms. Additionally, gene expressions do not always co-relate with protein expressions. We added a description in Discussion (Page 19).

4. The authors mentioned in the Introduction and Discussion sections that PSEN1 is a Ca2+ pore. However, they did not cite the most relevant papers, which directly showed conductive properties of PSEN1 using the BLM method (DOI: 10.1074/jbc.M111.243063; 10.1016/j.Cell.2006.06). These works should be cited.

Response: The manuscript [PMID: 21531718; DOI: 10.1074/jbc.M111.243063] was cited in the introduction. I was wondering what the other one would be. Did the reviewer mean PMID: 16923388 (DOI: 10.1016/j.cell.2006.06.044)?

MINOR issues

1. Fig. 2: I recommend to show also representative images of Cal-520 filled cells.

Response: The cell images filled with Cal-520 were added in the figure (New Fig 1). 

2. Fig. 6: The resolution of this figure has to be enhanced because labelling is not legible in the submitted version.

Response: We revised Fig 6, and Fig 6B was removed based on the comments by Reviewer 2 (Page 16). 

Reviewer #2: This work evaluated the role of ryanodine and IP3 receptors in the generation of calcium signals generated in cell lines derived from patients with familial forms of Alzheimer's disease with a mutation in presenilin 1. Although the results are of interest and can contribute to the discussion involving these receptors in Alzheimer's disease, several issues should be addressed before publication:

Response: We appreciate your time and efforts to review our manuscript and providing us with your valuable comments to improve our manuscript.

1) For example, the authors mention that" human neurons express RyR1 and RyR2" and ignore the RyR3 isoform, which is a brain isoform. They use inhibitors for RyR1 and RyR2 and conclude that RyR2 is very important, but they did not discuss the possible implications of RyR3. Moreover, in the hippocampal neurons, a brain region centrally involved in AD, low levels of RyR1 are present, while RyR3 is highly expressed.

Response: As the reviewer pointed out, RyR3 is expressed in human brains [https://www.ncbi.nlm.nih.gov/gene/6263]. However, dantrolene inhibits RyR3 as well as RyR1 [PMID: 11278295; PMID: 36982484]. Though this does not change our conclusion, the descriptions were added (Pages 4 and 19).

2) They also mentioned a study using dantrolene, an inhibitor of RyR1, as capable of eliminating both Aß42 and Ca2+ signals. However, they dismiss these findings, mentioning that these data still support the classical amyloid theory, which may not be the primary pathological mechanism. It is important to note that although a discussion has been made on the centrality of amyloid to AD pathogenesis, the amyloid cascade is still the leading hypothesis to explain the disease.

Response: The sentence, “These data still support the classical amyloid theory”, was removed.

3) Still on this matter, the authors omit a critical paper (https://doi.org/10.3389/fnmol.2017.00115) showing that Ca2+ release mediated by the RyR2 isoform causes the deleterious effects of AβOs on mitochondrial function in hippocampal neurons. Knockdown of RyR2 decreased by 40% Ca2+ release induced by the RyR agonist 4-chloro-m-cresol and significantly reduced the cytoplasmic and mitochondrial Ca2+ signals and the mitochondrial fragmentation induced by AβOs. Also, in another work from the same group (10.3389/fnagi.2018.00399), it was shown that AβOs injections directly into the hippocampus decreased RyR2 protein content but increased single RyR2 channel activation by Ca2+ and caused considerable spatial memory deficits. These articles should be cited and discussed correctly, together with Yao'paaper, in the manuscript.

Response: The aim of this study is to investigate whether PSEN1 variants found in early-onset AD can promote abnormal Ca2+-signals in human iPSC-induced neuronal cells. In other types of AD, AβOs can increase intracellular Ca2+ leading to neuronal damage. We modified the discussion section (Page 18-19). 

3) Figure 1 is a characterization of the ability to generate neurons from cell lines with previously described protocols, so it should be considered a supplementary figure. The figure legends do not specify the rum or the statistical test used.

Response: The initial Fig 1 was changed as Fig S1. These are representative neuronal images, and statistical methods were not used.

4) Regarding Figure 2, the authors mention that Ca2+-bursts were significantly higher in neurons bearing PSEN1 variants (p.A246E and p.M146L) compared to the WT neurons, suggesting that these two variants may cause more severe phenotypes than p.L286V. Although they show a quantification, no statistical analysis was presented to conclude that Ca2+ bursts were higher in those cases than the others.

Response: Statistical analysis was performed and shown as follows:

Comparisons of the rates of Ca2+-oscillations and Ca2+-bursts in neurons bearing PSEN1 variants were performed between p.A246E and the WT neurons, between p.M146L and the WT neurons, and between p.L286V and the WT neurons. Regarding Ca2+-oscillations, the 99% confidence intervals (CI) of these comparisons were all significant. (Table S1) Comparisons of the rates of Ca2+-bursts were [99%CI: -0.483, -0.338] (p.A246E and WT) and [99%CI: -0.224, -0.020] (p.M146L and WT), respectively, both of which were significant (p<0.01). Meanwhile, a comparison of the rates of Ca2+-bursts in neurons with PSEN1 variants between p.L286V and the WT neurons showed [95%CI: -0.100, 0.019] which was nonsignificant. (Table S2) Therefore, we suggested that the two variants with p.A246E and p.M146L may cause more severe phenotypes than p.L286V (Page 12). Accordingly, Fig 2 was revised.

5) In Figure 3, the results are coherent, but what were the controls to check that BAPTA and CPA did not kill the cells after their appliance to the cultures?

Response: In the revised Fig 3, we added the traces obtained from the same cells after washout of BAPTA and CPA. As you can see, those cells showed Ca2+-bursts after washout (Page 13-14).

6) In the figure 4, they used carvedilol to block RyR2. Using genetic tools to imply RyR2 in the Ca2+ signals would be relevant. Also, although n is shown for each condition in the figure legend, statistical analysis needs to be mentioned, which is needed to make conclusions about the role of RyR2 as the "main facilitator of the abnormal Ca2+-bursts in the cells bearing PSEN1 variant.". Moreover, significant differences exist among the N used in the different conditions. They mention N = 21 in B, N = 98i C, and N = 18 in D. Given this disparity and the elevated values for N, it is necessary to describe what the authors consider an N.

Response: We randomly chose neuronal cells showing Ca2+-bursts, and the effects of the reagents were consistent in all cells (i.e., all-or-none). We did not intentionally emphasize the role of RyR2 based on these numbers. In general, approximately 20 consistent observations are thought to be a true phenomenon (i.e., not experimental error). Please see some of our previous papers for examples [PMID: 12208804; PMID: 15767295]. Descriptions were added on Page 14.

7) The authors should describe how they performed quantifying the immunofluorescence images in Figure 5.

Response: Based on the other reviewer, this data was replaced with a Western blot data. Please see the revised methods and results section (Page 15).

8) RyR3 was omitted until Figure 6, where it appears to be decreased just as RyR2, but without any comment. Also, the scheme shown in Figure 6B is speculative and can not be supported by the results presented in this work. Moreover, the resolution is inacceptable in this figure.

Response: Fig 6B was removed based on the reviewer’s comment and Fig 6A was revised (New Fig 6).

---

## [Decision Letter · Decision Letter 1]

2 May 2024

Carvedilol suppresses ryanodine receptor-dependent Ca2+ bursts in human neurons bearing PSEN1 variants found in early onset Alzheimer’s disease

PONE-D-23-23711R1

Dear Dr. Ai,

We’re pleased to inform you that your manuscript has been judged scientifically suitable for publication and will be formally accepted for publication once it meets all outstanding technical requirements.

Kind regards,

Pan Li, PhD

Academic Editor

PLOS ONE

Additional Editor Comments (optional):

Reviewers' comments:

Reviewer's Responses to Questions

**Comments to the Author**

1. If the authors have adequately addressed your comments raised in a previous round of review and you feel that this manuscript is now acceptable for publication, you may indicate that here to bypass the “Comments to the Author” section, enter your conflict of interest statement in the “Confidential to Editor” section, and submit your "Accept" recommendation.

Reviewer #1: All comments have been addressed

2. Is the manuscript technically sound, and do the data support the conclusions?

Reviewer #1: Yes

3. Has the statistical analysis been performed appropriately and rigorously? 

Reviewer #1: Yes

4. Have the authors made all data underlying the findings in their manuscript fully available?

Reviewer #1: Yes

5. Is the manuscript presented in an intelligible fashion and written in standard English?

Reviewer #1: Yes

6. Review Comments to the Author

Reviewer #1: The authors addressed all issues and made appropriate changes in the revised manuscript. The paper is now ready for publication.

7. PLOS authors have the option to publish the peer review history of their article (what does this mean?). If published, this will include your full peer review and any attached files.

Reviewer #1: **Yes: **Marta Gaburjakova, PhD

---

## [Editor Report · Acceptance letter]

15 Jul 2024

PONE-D-23-23711R1 

PLOS ONE

Dear Dr. Ai, 

I'm pleased to inform you that your manuscript has been deemed suitable for publication in PLOS ONE. Congratulations! Your manuscript is now being handed over to our production team.

Kind regards, 

on behalf of

Dr. Pan Li 

Academic Editor

PLOS ONE